# Sex and *APOE* ε2 Interactive Effects on the Longitudinal Change in Cognition in a Population-Based Cohort of Older Adults with Vascular Risk Factors

**DOI:** 10.3390/ijms262110591

**Published:** 2025-10-30

**Authors:** Noemí Lamonja-Vicente, Rosalía Dacosta-Aguayo, Jorge López-Olóriz, Laia Prades-Senovilla, Juan José Soriano-Raya, Inmaculada C. Clemente, Júlia Miralbell, Maite Barrios, Elena López-Cancio, Cynthia Cáceres, Mónica Millán, Pere Torán, Guillem Pera, Meritxell Carmona-Cervelló, Cecilia Herrero, Pilar Montero-Alia, Maria Palau-Antoja, Maria Hernández-Pérez, Tamara Canento, Ana Gonzalez Fuxa, Maria Mataró, Marc Via

**Affiliations:** 1Department of Clinical Psychology and Psychobiology, Universitat de Barcelona, Passeig de la Vall d’Hebron 171, 08035 Barcelona, Spain; rdacostaa@gmail.com (R.D.-A.); laia5.3032@hotmail.com (L.P.-S.); juan.jose.soriano@ub.edu (J.J.S.-R.); iclemente@ub.edu (I.C.C.); juliamiralbell@gmail.com (J.M.); mmataro@ub.edu (M.M.); 2Institut de Neurociències, Universitat de Barcelona, 08035 Barcelona, Spain; mbarrios@ub.edu; 3Unitat de Suport a la Recerca Metropolitana Nord, Fundació Institut Universitari per a la Recerca a l’Atenció Primària de Salut Jordi Gol i Gurina, 08303 Mataró, Spain; ptoran.bnm.ics@gencat.cat (P.T.); gpera.bnm.ics@gencat.cat (G.P.); mcarmonace.mn.ics@gencat.cat (M.C.-C.); cherreroal.mn.ics@gencat.cat (C.H.); pmonteroalia@gmail.com (P.M.-A.); mariapalauantoja@gmail.com (M.P.-A.); 4Faculty of Health Sciences, International University of la Rioja (UNIR), 26006 Logroño, Spain; 5Centre de Serveis de Psicologia UB (CSPUB), Fundació Josep Finestres (FJF), 08035 Barcelona, Spain; lopezjorge84@hotmail.com; 6Institut d’Investigacions Biomèdiques August Pi i Sunyer (IDIBAPS), 08036 Barcelona, Spain; 7Institut de Recerca Sant Joan de Déu, Esplugues de Llobregat, 08950 Barcelona, Spain; 8Department of Social Psychology and Quantitative Psychology, Universitat de Barcelona, 08035 Barcelona, Spain; 9Unidad de Ictus, Departamento de Neurología, Hospital Universitario Central de Asturias (HUCA), 33011 Oviedo, Spain; elenacancio@gmail.com; 10Department of Neuroscience, Hospital Universitari Germans Trias i Pujol, 08916 Badalona, Spain; caceresa@hotmail.com (C.C.); mhernandez@igtp.cat (M.H.-P.); canento88@gmail.com (T.C.); anagon14_2@hotmail.com (A.G.F.)

**Keywords:** *APOE*, *BDNF*, sex, longitudinal, cognition, aging

## Abstract

Cognitive aging trajectories differ widely across individuals, and genetic factors such as *APOE* and *BDNF* polymorphisms may contribute to this variability. While *APOE* ε4 has been widely studied, the influence of *APOE* ε2, particularly in interaction with sex, remains underexplored. This study aims to examine the longitudinal trajectory of *APOE* ε2 individuals on cognitive performance, and their interactions with sex, age, and *BDNF* Val66Met polymorphism, in a population-based cohort of older adults with vascular risk. We analyzed data from 386 participants (mean age: 71.8) from the Barcelona-AsIA Neuropsychology Study, followed over a 7-year period. Verbal memory, verbal fluency, and visuospatial domains were assessed. Linear regression models tested associations between cognitive change and genotypes, controlling for age, sex, education, depression, and vascular risk. Interaction terms and permutation testing were applied. Regression to the mean (RTM) effects were assessed. *BDNF* showed no significant associations with cognitive performance. RTM effects were evident across subgroups, particularly among ε2 carriers, suggesting this phenomenon partly explains the divergent results over time. *APOE* ε2 does not confer a consistent protective effect on cognition over time. Our results highlight that *APOE* ε2 may be detrimental to verbal memory in aging males.

## 1. Introduction

Most cognitive abilities decline with age, affecting the ability to perform basic daily functions and leading to a reduced sense of well-being [1,2]. Nevertheless, interindividual differences are considerable and not uniform throughout the lifespan. The identification of the factors involved in cognitive decline processes in late life is of utmost relevance for the development of effective and more personalized interventions [3].

Among the various genetic factors involved, *Apolipoprotein E* (*APOE*) plays a pivotal role in the risk of Alzheimer’s disease (AD) [4,5,6]. The functional impact of the *APOE* alleles—ε2, ε3, and ε4—has long been considered within a framework that ranges from beneficial to adverse across the life course, affecting cognition distinctly during different life stages [7,8]. The ε4 allele has been extensively examined due to its role as a genetic risk factor for cognitive decline in individuals with normal aging, mild cognitive impairment, and Alzheimer’s disease [9,10]. In contrast, the ε2 allele has been traditionally associated with enhanced cognitive performance across various domains in aging individuals, as well as a slower rate of age-related cognitive decline in both healthy and cognitively impaired populations [11,12,13,14]. However, emerging research suggests that its impact on cognition is inconsistent and appears to be limited [15]. The comprehensive review conducted by Kim et al. concluded that the evidence for the cognitive benefits of ε2 is mixed, and methodological limitations may have contributed to misleading conclusions in some prior studies [15]. In another large-scale observational study conducted by Wood et al., the authors revealed sex-specific effects, showing that among non-Hispanic White adults, ε2 carriers experienced slower cognitive decline only in men, but not in women [16]. Overall, ε2 is still linked to a lower risk of Alzheimer’s disease and longer lifespans, but its effects on cognition, especially in healthy aging, are not all equally positive. Recent research indicates that these effects vary by sex, ethnicity, and phenotype. Even so, it remains the case that studies focused on the role of ε2 in cognition have attracted less attention in comparison with studies based on the role of ε4 in cognition due mainly to the limited frequency of the ε2 variant (7% in the global community [17]) and the design of studies in which ε2 carriers are either excluded or combined with ε3 in a non-ε4 group. Although *APOE* is a key genetic factor for AD and cognition, its effects vary across individuals, and several factors have been proposed as modulators of this heterogeneity.

One of the primary variables that can influence how *APOE* affects cognition is sex. Its influence on the association between the *APOE* ε4 allele, cognitive function, and Alzheimer’s disease risk has been highlighted in several studies [4,18,19,20]. Women are at higher risk of developing late-onset AD (LOAD), probably due to an intensification of the effects of the loss of sex hormones during peri-/post-menopause [21,22]. The female brain may be more susceptible to cognitive decline and neurodegeneration as a result of fluctuating estrogen levels during perimenopause and the marked drop in estrogen following menopause [23,24,25]. In fact, brain glucose hypometabolism, Aβ deposition, and loss in brain volume commence at perimenopause [26,27,28,29]. Numerous studies have documented a more pronounced negative impact of the *ε4* allele in women across a range of Alzheimer’s disease biomarkers [30,31,32,33,34,35]. Only two studies centered on the protective effects that *APOE ε2* seems to have in females [36,37]. The first one, conducted by [36], was a 4 to 7-year longitudinal study in which the authors reported a more pronounced protective effect of *APOE ε2* among women in predicting performance on a delayed recall task. The second one, conducted by [37], was a Genome-Wide Association study in which the authors reported a significant association between *APOE ε2* and slower cognitive decline in women. Although those studies showed sex differences in the effects of *APOE* on cognition, few studies to date have explicitly examined the interactive effects of sex**APOE* on cognitive performance. In our previous cross-sectional study [38], we reported for the first time a significant interaction between sex**APOE ε2* in females in the verbal memory and verbal fluency domains compared to men. To the best of our knowledge, only one longitudinal study explicitly examined *APOE* × sex interactions [39], and they found no significant effects of sex × *APOE ε4* on cognitive performance. More recent longitudinal studies, however, have started to provide insight into this subject. For example, Burns et al. (2025) found that *APOE ε4* status and sex have differential impacts on the brain’s longitudinal dynamics of excitation–inhibition balance. This differential impact is discussed as a preclinical indicator of cognitive decline in female *APOE ε4* carriers. Further evidence for sex-modulated effects of *APOE* on cognitive outcomes comes from Dessy et al. (2024), who looked at the disentangled effects of gender and sex on *APOE ε4*–related neurocognitive impairment [40,41]

Longitudinal studies reported that ε4 carriers had an increased risk of developing mild cognitive impairment (MCI) [42], AD [43], and vascular dementia [44]. Less abundant literature has investigated the effects of *ε2* on cognition, suggesting that *ε2* carriers exhibit moderately preserved cognitive function over time. In young adults, it has been shown that *APOE ε2* carriers had better spatial strategies compared to *ε3* and *ε4* carriers [45]. In older adults, *APOE ε2* carriers have shown a protective effect on a measure of global cognitive function [46], and a reduced risk and delayed onset of AD [47,48,49]. Other studies question the presence of such benefits, even suggesting a cognitive disadvantage [50,51,52], or no effect of *ε2* on cognitive change ratios [53,54].

Other potential factors that may interact with the effect of *APOE* on cognition include lipidic profile, inflammation, interactions with polymorphisms in other genes—such as *Sponding-1 (SPON1)* rs11023139 [55] or the *Brain-derived neurotrophic factor (BDNF) Val66Met* polymorphism [56,57]—as well as dietary protein intake [58], having positive or negative age beliefs [59], physical activity [60,61], or education [62].

Special attention deserves the interaction between the *APOE* and the *BDNF*. *BDNF* is a key player in synaptic plasticity, long-term potentiation, and hippocampus-dependent memory, and it is necessary for normal cognitive function. According to new research, *APOE* genotype and *BDNF* interact, specifically affecting cognitive outcomes over time. The *APOE*-ε4 allele significantly altered the association between the *BDNF* Val66Met polymorphism and cognitive impairment, suggesting an interactive risk profile in nonclinical aging, according to Ji et al.’s (2024) study of older adults living in the community. Vilor-Tejedor et al. (2020) demonstrated that the *BDNF Met* variant compensates for *APOE-ε4*-related decreases in hippocampal subfield volumes in middle-aged, cognitively unimpaired individuals from the ALFA cohort, indicating structural resilience mechanisms associated with genotype interaction. Edmunds et al. (2025) demonstrated that circulating *BDNF* expression mediates verbal learning and memory in a cohort enriched for Alzheimer’s risk. [63,64,65]. All of these studies demonstrate the importance of *BDNF* in supporting cognitive health and reducing the risk of *APOE*, particularly in female subgroups and prior to the onset of clinical symptoms. To the best of our knowledge, there is no previous longitudinal research studying the interaction between sex and *APOE* ε2 on cognition, nor the interaction between *BDNF* and *APOE*. This type of research is crucial for understanding the mechanisms underlying the protective and deleterious effects of *APOE* on cognition over time. This study aims to analyze the effects of the *APOE* genotype on changes in cognitive performance over seven years, as well as its interaction with sex, age, and *BDNF* genotype. Based on our previous results [38], we anticipate that the *ε2* allele will be associated with a lower rate of cognitive decline, particularly in verbal memory and fluency domains, and among female participants. Additionally, we anticipate that the combination of *APOE ε4+* and the *BDNF* Met allele will be significantly associated with worse memory performance compared to other genotypes, and that *APOE ε2*, as a positive regulator of *BDNF*, may have a protective effect [63,66,67].

## 2. Results

### 2.1. Subjects

The initial sample, which underwent a complete neuropsychological (NPS) assessment between 2007 and 2010, included 747 subjects, of whom 648 had *APOE* genotyping. Of these, a total of 386 were accepted as being cognitively reassessed between April 2016 and May 2017. (Figure 1).

The mean age of the final sample was 71.82 years (SD = 6.58). Moreover, the 36.80% of the participants were females (n = 142) and reported 6.76 (SD = 4.22) years of education on average. Summary details of the sample are described in Table 1.

Sociodemographic characteristics comparison between subjects who only completed the NPS baseline evaluation (n = 260) and subjects with baseline and follow-up assessments (n = 386) showed that subjects that did not undergo a new complete NPS assessment were older (t (648) = −5.07; *p* < 0.01), with less education years (t (648) = 2.70; *p* = 0.01) and lower cognitive performance (*p* < 0.01 for all three domains) and with higher levels of vascular risk factors based on the REGICOR score (t (648) = −3.23; *p* < 0.01) than subjects who were reassessed in the follow-up (Appendix A). Conversely, there were no differences in allele frequency between these subgroups.

### 2.2. Genotype and Cognitive Domains

Our primary goal was to investigate the potential associations between the genetic variants analyzed and changes in cognitive performance. When we analyzed cognitive change (i.e., follow-up minus baseline difference), we observed a higher decline in verbal memory performance among ε2-carriers, especially among male participants (Table 2). After permutations, results remained significant only among male participants (β = −0.382; 95% CI: −0.670/−0.095; *p* = 0.01; *p*-perm = 0.027). Women showed no evidence of such an effect. Similar effects were observed when comparing participants carrying the *APOE* ε2 allele (ε2/ε2 and ε2/ε3) to those with the ε3/ε3 genotype, excluding ε4 carriers.

Regarding the *APOE* ε4 allele, no significant associations were found between changes in performance across any of the cognitive domains studied and the APOE ε4 allele, after adjusting for covariates (Table 2).

The associations observed between the *APOE* ε2 allele and verbal memory performance were confirmed in the subgroup of participants without depressive symptoms (GDS > 5, N = 303), thus ruling out potential confounding effects due to depression.

### 2.3. Cross-Sectional Analyses

These initial longitudinal findings appeared to be discrepant with our hypotheses and previously published cross-sectional results from the same cohort. In our early study, we found that ε2-carriers performed significantly better in the verbal memory and fluency domains at baseline, and these results were particularly notable among women [38]. To assess whether differences between participants with only the baseline assessment and those with both baseline and follow-up assessments might account for these discrepancies, we ran association analyses of cognitive performance in the baseline assessment separately for each subgroup. Results matched our previously published findings in both subgroups separately: a significant protective association of the ε2 allele on the verbal memory and fluency domains among women at baseline (Appendix A).

At follow-up, ε2 carriers showed a similar cognitive performance to that of their non-carrier counterparts. The protective effects of ε2 found at baseline were not present in the follow-up assessment, and there were no significant cognitive differences between *APOE* genotypes (Table 3). We observed an adverse trend effect among males (*p* = 0.056) and a protective trend effect among females (*p* = 0.073) in the verbal memory domain, although these observations did not reach statistical significance. The other variants studied in the *APOE* gene did not show relevant results in the association analyses on cognitive performance at follow-up.

#### Regression to the Mean

In order to reconcile the apparently conflicting findings of the effect of the ε2 allele on verbal memory (a protective role among women at baseline, no effect at follow-up, and a detrimental effect on longitudinal change among men), we explored whether a potential regression to the mean (RTM) effect could contribute to our observations of the longitudinal effects. Examination of the data revealed RTM effects in all subgroups (Figure 2). Overall, individuals whose baseline results in the verbal memory domain were higher than the average tended to perform worse in the follow-up measurement and cognitive changes were likely to retrieve negative values (lower right side of the plot). The inverse pattern was observed for individuals with low baseline scores. In addition, there were differences in RTM effects depending on ε2-carrier status and sex. In the whole sample, the intensity of RTM effects was seemingly similar (i.e., regression lines with the same slope) in ε2 carriers and non-carriers (Figure 2A), with a trend towards more negative values in cognitive change among ε2 carriers as revealed in the previous analyses. However, we observed that the effects were different among ε2 carriers and non-carriers when stratifying by sex (Figure 2B,C). Individuals carrying ε2 alleles showed more intense RTM effects compared to their non-carrier counterparts in both male and female subgroups, as illustrated by steeper slopes of the regression lines.

We did include cognitive performance in the baseline evaluation as a covariate in longitudinal analyses to account for potential RTM effects. This covariate was highly significant in most of the regression models, an indication that it successfully corrected (at least partially) biases induced by extreme baseline values. When we repeated analyses in the verbal memory domain, adjusting by the mean between the first and second questionnaires’ scores as an alternative method to capture RTM effects, we retrieved very similar results, with adverse effects of the ε2 allele from baseline to follow-up (i.e., higher memory decline) in the whole sample and among males (Appendix A).

### 2.4. Gene*Age and Gene*Gene Interactions

We further explored the effect that gene-age and gene-gene interactions could play in the evolution of cognitive performance during the aging process. Regarding interactions with age, we did not identify significant interactive effects in the regression models (*p* > 0.3). However, a stratified analysis revealed that the detrimental effects of the ε2 allele on longitudinal change in verbal memory were only present among the younger males (i.e., younger than 71 years old, the median age in our cohort) (Appendix A).

Since our previous analyses in the baseline assessment revealed significant *APOE***BDNF* interactions on cognitive performance, we also explored the presence of gene-gene interactive effects on cognitive decline in the longitudinal dataset. However, we did not find significant results (*p* > 0.05, Appendix A). Moreover, similarly to our previous observations at baseline, *BDNF* did not show any significant association, as an independent factor, with the cognitive domains at either follow-up or the longitudinal level.

## 3. Discussion

*APOE* is widely recognized as a significant factor in cognitive success or decline during the aging process. Nevertheless, the complexity of its genotype interactions with age, sex, and other variables, such as vascular risk factors, hormones, and *BDNF*, remains a matter of controversy [68]; the mechanisms through which it exerts its effects also remain unclear. In this paper, we investigated the effects of genetic variants in the *APOE* gene on the longitudinal change in cognition in a large cohort of community-dwelling older adults. We focused on studying the interactive effects of sex, the *BDNF* Met allele, and *APOE* ε2 on cognition. Our results showed a detrimental association of *APOE* ε2 in males with changes in verbal memory, whereas no significant protective impacts were found for females in any cognitive domain. We did not find any influence of age or *BDNF* in our results.

Research centered on the role of *APOE ε2* on cognitive decline is scarce, and the results are mixed. Some studies have reported a positive effect of *APOE ε2* on reducing cognitive decline in both healthy and demented populations [13,46,69,70], as well as improved performance in short-term and long-term memory [71,72,73,74] and executive function and attention [13]. Other studies have questioned the presence of such benefits, even suggesting a cognitive disadvantage [50,51,52,75]. None of these studies, however, focused on the differential effect of sex on their results. Only a few studies have focused on the differential effect of *APOE ε2* on cognition by sex. While one study reported a protective effect of *APOE ε2* among women in predicting performance on a delayed recall task [36], another observed a protective effect of *APOE ε2* against cognitive decline in men [16].

In our previously published cross-sectional study [38], we reported a significant interaction between sex and *APOE ε2* on cognitive performance in females. Specifically, we found that female *APOE ε2* carriers performed better in the domains of verbal memory and verbal fluency and that total cholesterol mediated these protective effects on cognition, providing a physiological mechanism for the observed genetic effects. In the present longitudinal study, and contrary to our expectations, the effect of the *APOE* ε2 genotype was deleterious for males in the verbal memory domain compared to non-ε2 carriers. That is, male ε2 carriers experienced more cognitive decline than non-ε2 carriers, finding evidence of an adverse effect of *APOE ε2* on cognition in males. We did not, however, find the expected protective effect of *APOE ε2* in females in the verbal memory and verbal fluency domains. This result can be explained by the differential characteristics found between subjects who received a second assessment and those who dropped out of the study. Subjects who dropped out of the study were older, less educated, had lower cognitive performance, and exhibited higher levels of vascular risk factors compared to the subjects who were followed up in the study. In fact, when we run separate cross-sectional analyses for subjects with both assessments and participants with only the baseline assessment, our main results match our previously published findings: a significant protective effect of the *APOE* ε2 allele on the verbal memory and fluency domains among women in both groups. It is possible that we retained subjects in our sample who performed better cognitively and had a better lifestyle, considering they had lower levels of vascular risk factors compared to the participants who dropped out. When we conducted a cross-sectional analysis at the follow-up visit (seven years after the initial assessment), we did not find significant effects of the ε2 allele; however, we observed trends towards a protective effect in women and a detrimental effect in men. Furthermore, although we controlled for the RTM effects, the fact that the RTM effects varied depending on ε2-carrier status and sex could have contributed to the apparent discrepancy between our previous study and this one. *APOE* ε2 carriers showed more intense RTM effects than non-carriers, separately in both the male and female subgroups.

Regarding age, one potential explanation for the lack of significant effects of age on the *APOE* genotype status could be that our age range is limited to 13 years. In contrast, studies focused on the effect of age on *APOE* ε2 carriers considered several decades for their analysis [76].

Several factors, beyond age and sex, may be interacting or mediating the effect of *APOE* on cognition and could serve as a putative explanation for the results found in our present study. Among them, the lipidic profile stands out as particularly relevant. *APOE* is involved in the transport of lipids in the body and brain [77]. It is known that *APOE* ε4 is associated with higher levels of low-density lipoprotein cholesterol (LDL-C) and total cholesterol (TC), as well as an increased risk of dementia [78]. In contrast, *APOE ε2* is associated with elevated very low-density lipoprotein (VLDL) and a lower risk of dementia [79]. Women typically exhibit higher levels of high-density lipoprotein cholesterol (HDL-C) and lower levels of LDL-C and VLDL compared to age-matched men [80]. However, over time, the TC and LDL-C profiles become less favorable in women with lower age-specific anti-Müllerian hormone levels, a hormone related to ovarian reserve [81]. This would explain, for example, the higher risk of cerebrovascular disease factors in women at menopause [81]. It is known that cholesterol accumulation leads to subsequent neuroinflammation and neurodegeneration, leading to cognitive dysfunction. In our previous study, we found that TC mediated the protective effect of *APOE* ε2 on cognitive performance in females, suggesting that sex differences should be considered when analyzing the effect of specific genes on cognition. Unfortunately, we did not have available data on lipid levels at follow-up, and therefore, we cannot analyze their mediator effect as we did at baseline. This is a significant limitation in our study that prevents us from further developing a putative explanation regarding the change in the lipidic profile in women and how this change could interact with *APOE* ε2 and cognition.

Sexual dimorphism in brain function is driven by complex biological pathways involving sex hormones. Females generally exhibit higher cerebral blood flow (CBF) than males—a difference that persists with aging despite the overall age-related decline in CBF [82,83,84]. Most studies attribute this advantage to estrogens, which have well-established neuroprotective and vasoregulatory roles [85]. In contrast, the impact of androgens on neuroinflammation and cerebrovascular regulation is less understood. Evidence suggests that androgens can exert both protective and detrimental effects on the cerebral vasculature [86]. Furthermore, androgen levels decline from early adulthood due to reduced testicular function, contributing to oxidative stress, diminished synaptic plasticity, and cognitive decline [87,88].

Other factors that should be considered as potential modulators of the observed *APOE**sex effects include *BDNF* [56,89], dietary protein intake [58], having positive or negative age beliefs [59], education [62], physical activity [60,61], and air pollution [90]. In particular, at baseline, we previously found a significant interaction between genotypes in the *APOE* and *BDNF* genes [38] that has not been replicated in our present longitudinal work. Regarding physical activity, for example, we have recently reported that higher cardiorespiratory fitness (CRF) mediated the cognitive benefits of physical activity on executive function and attention-speed domains only in men but not in women [91]. Moreover, although we did not collect information on environmental exposures, increasing evidence points to a relevant effect of air pollution on cognition, and some of its effects appear to be sex-specific [90].

Considering all the aforementioned potential limitations, the results from the present study were consistent with a protective effect of the *APOE ε2* allele on verbal memory, especially among women, which diminished with age. That would explain the protective effect at baseline, the trend at follow-up, and the absence of a longitudinal effect. Among men, the protective effect was absent or substantially smaller, and a combination of biases in the cognitive profile of dropouts (i.e., participants in the longitudinal phase presented higher cognitive performances at baseline) and RTM effects could explain the apparently adverse longitudinal effects of ε2.

## 4. Materials and Methods

### 4.1. Sample/Participants

The Barcelona-AsIA (Asymptomatic Intracranial Atherosclerosis) Neuropsychology Study is a longitudinal, prospective population-based study [47]. The study was conducted from March 2007 to May 2017 at the Hospital Universitari Germans Trias i Pujol in Badalona, Spain. Briefly, 933 participants over 50 years of age, with a moderate–to–high vascular risk in REGICOR [92], were recruited. REGICOR is a version of the Framingham risk score adapted and validated for the Spanish population, which includes age, sex, smoking, diabetes, TC, HDL-C, and blood pressure to estimate vascular risk. Exclusion criteria included dementia (Mini-Mental State Examination score < 25), history of stroke or transient ischemic attack, coronary heart disease, chronic neurological disease, psychiatric disorder, severe disability or institutionalization, and other medical conditions that could affect cognitive assessment and function. Recruitment protocol details have been previously reported [93,94]. Participants underwent a baseline assessment between 2007 and 2010, which included cognitive and behavioral assessments, as well as the extraction of blood samples. After different quality controls, 648 individuals had baseline data on genotype and neuropsychology (see [38] for further details). At follow-up, participants underwent cognitive reassessment between April 2016 and May 2017. The study was approved by the Germans Trias i Pujol University Hospital Ethics Committee (IRB00002131). All participants gave their written consent to participate in the study.

### 4.2. Neuropsychological and Behavioral Assessment

Participants were assessed using a comprehensive neuropsychological battery at the baseline and 7 years later in the follow-up visit. Performance on each test (direct scores) was standardized to Z-scores, separately for each visit. Factorial analysis from baseline data previously published [95] generated three cognitive domains: (a) Visuospatial skills/speed, (b) Verbal memory, and (c) Verbal fluency (Appendix A). Cognitive change in the longitudinal study was calculated as the difference in the Z-scores between visits (follow-up minus baseline). Depression was assessed at both time points with the Short Geriatric Depression Scale (GDS-15), with scores >5 considered indicative of potential depression [96].

### 4.3. Genetic Analyses

Extensive details of the genetic analyses have already been published [20]. Briefly, blood samples were obtained following an overnight fast and stored in a biobank at −80 °C until DNA was extracted using an ISOLATE II Blood DNA Kit (BIOLINE, London, UK). *APOE* (rs429358 and rs7412) and *BDNF* (rs6265) SNPs were genotyped using KASPar assays by an external facility (Progenika Biopharma S.A., Derio, Spain). Negative controls and sample duplicates were included for quality control. *APOE* genotypes from the two SNPs were recoded into the classical ε2/ε3/ε4 allele classification following the standard nomenclature. We removed samples with more than one missing genotype, and all SNPs in our sample were in Hardy–Weinberg equilibrium.

### 4.4. Statistical Analyses

We used an independent sample *t*-test to analyze sociodemographic and cognitive differences between subjects who completed both the baseline and follow-up assessments and subjects who were lost to follow-up and did not complete the follow-up assessment. Demographic and clinical data were analyzed using the Statistical Package for Social Sciences version 24 (SPSS Inc., Chicago, IL, USA).

To examine the effects of *APOE* genotypes on longitudinal cognitive change, we performed linear regression analyses on the difference scores (follow-up minus baseline) for each cognitive domain. Covariates included age, sex, years of education, depressive symptoms, and cardiovascular risk (as measured by the REGICOR score). Baseline cognitive performance was included as a covariate to reduce potential bias and account for regression-to-the-mean effects. Significant results were further tested using permutation procedures (1000 permutations) to confirm robustness and reduce the risk of Type I error. To evaluate cross-sectional associations between *APOE* genotypes and cognition, we conducted separate linear regression analyses at baseline and follow-up. We also conducted stratified analyses based on depressive symptoms (GDS > 5) to identify potential confounding effects on the observed associations. These analyses also included the same covariates. All genetic association analyses (longitudinal and cross-sectional) were performed using PLINK 1.9 ([52]; www.cog-genomics.org/plink/1.9/ accessed on 1 February 2024) under additive genetic models. *APOE* genotype was modeled using comparisons of ε2 vs. non-ε2 and ε4 vs. non-ε4 alleles. Interaction effects between *APOE* genotypes, *BDNF*, sex, and age were also explored.

We examined the presence of ‘regression to the mean’ (or ‘regression towards the mean’, RTM) effects in our observations to control for potential biases that can affect analyses of longitudinal data. RTM is a phenomenon that may arise when the value of a random variable is extreme (either low or high) at a sampling point; a future measurement of the variable would be closer to the mean [97]. RTM effects become more noticeable when follow-up measurements are compared between sub-samples with differences in baseline values, as in our case, where ε2 carriers showed higher verbal memory scores at baseline. RTM effects should have been partially corrected in the association analyses, since longitudinal analyses were already adjusted for the baseline score. However, this crude adjustment might not be enough under specific scenarios of interindividual variability and measurement error [98]. We conduct post hoc analyses of the significant association results from the longitudinal study, adjusting for the mean difference between baseline and follow-up scores [99]. Moreover, examination for RTM effects included plotting cognitive score change (follow-up minus baseline values) over baseline scores, stratifying by ε2 carrier status and sex.

## 5. Conclusions

In summary, sex is a physiological factor that contributes to changes in various biological pathways, determining whether an individual will experience impaired cognition with aging, depending on their genetic background. However, studies specifically addressing these interactions remain scarce. We emphasize the importance of carefully examining the sex-specific effects of *APOE ε2* on cognitive decline and brain metabolism, structure, and function throughout the lifespan. Overall, our results are consistent with a protective effect of the *APOE ε2* allele, which is especially relevant among women. However, this effect appears to diminish with age, as previously noted in some studies [100]. This may explain the remarkable effect observed in our previous cross-sectional analysis at baseline in women, with effect sizes on verbal memory of 0.83 SD (95% CI: 0.49–1.18; *p*-perm = 10^−4^) per ε2 allele carried, and its attenuation at follow-up in the current dataset, where only a weak non-significant trend remained. In contrast, among men, the adverse longitudinal effect of ε2 on verbal memory may be partially explained by regression to the mean (RTM), especially given the absence of significant effects at baseline or follow-up. Importantly, this supposed detrimental effect was more negligible (0.38 SD; 95% CI: 0.09–0.67; *p*-perm = 0.03) than the protective effect observed in women at baseline. The differential effect of *APOE ε2* on sex warrants further investigation before firm conclusions can be drawn.

Notable limitations of this study include (1) a low number of participants with frequency of *APOE* ε2 alleles in the sample; (2) potential RTM effects, even after controlling for them; (3) sample loss between baseline and follow-up, with dropouts showing poorer cognitive performance, lower education level, and more vascular risk factors, especially in males; and (4) the lack of follow-up data on key modulators, such as lipid profile and cognitive reserve.

## Figures and Tables

**Figure 1 ijms-26-10591-f001:**
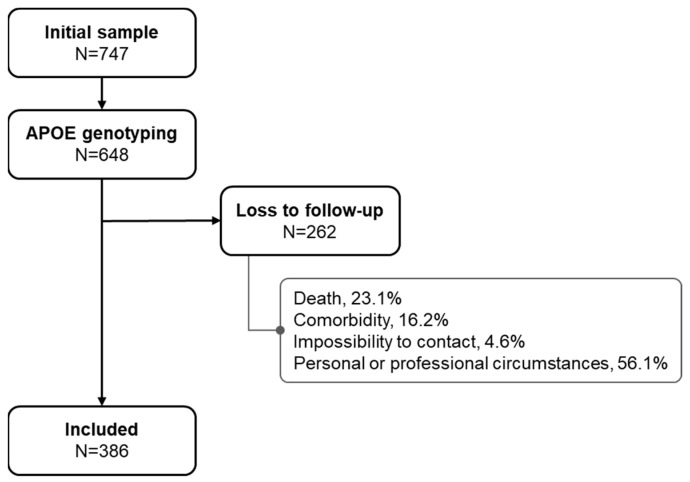
Flowchart of the study population.

**Figure 2 ijms-26-10591-f002:**
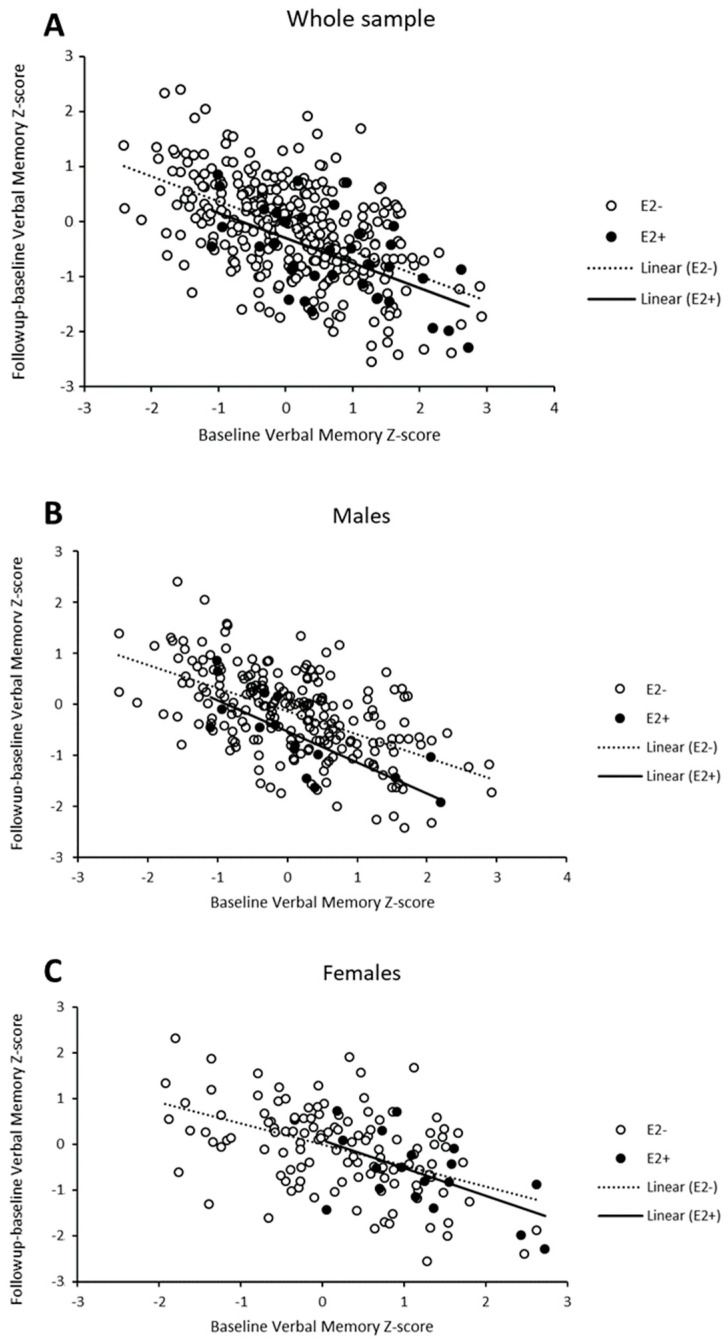
Scatter-plot of verbal memory Z-scores across time points showing change (follow-up minus baseline) against baseline scores for the whole sample (**A**) and stratified for males (**B**) and females (**C**).

**Table 1 ijms-26-10591-t001:** Sociodemographic, clinical, and neuropsychological sample characteristics.

	Baseline			Follow-Up		
	Total (n = 386)	Men (n = 244)	Women (n = 142)	Total (n = 386)	Men (n = 244)	Women (n = 142)
Sociodemographic and clinical data						
Age (years)	65.04 (6.83)	66.14 (6.95)	63.15 (6.20)	71.82 (6.58)	72.80 (6.82)	70.13 (5.79)
Education (years)	6.76 (4.22)	6.83 (4.61)	6.63 (3.46)	6.76 (4.22)	6.83 (4.61)	6.63 (3.46)
GDS-15	2.34 (2.56)	1.89 (1.99)	3.10 (3.18)	1.41 (2.26)	1.01 (1.71)	2.08 (2.83)
REGICOR	7.76 (3.58)	8.64 (3.83)	6.25 (2.46)	No data	No data	No data
ε2 vs. non-ε2 (% ε2 carriers (n))	9.8 (38)	7.8 (19)	13.4 (19)	9.8 (38)	7.8 (19)	13.4 (19)
ε4 vs. non-ε4 (% ε4 carriers (n))	17.9 (69)	20.1 (49)	14.1 (20)	17.9 (69)	20.1 (49)	14.1 (20)
Neuropsychological domains (Z scores)						
Visuospatial skills/speed	0.06 (0.42)	0.08 (0.42)	0.01 (0.41)	0.16 (1.35)	0.19 (1.54)	−0.02 (0.90)
Verbal memory	0.09 (0.92)	0.02 (0.94)	0.21 (0.87)	−0.01 (0.92)	−0.11 (0.89)	0.16 (0.95)
Verbal fluency	0.04 (0.84)	0.12 (0.89)	−0.11 (0.71)	0.10 (0.95)	0.17 (0.97)	0.00 (0.93)

Note: Continuous variables are reported as mean (SD).

**Table 2 ijms-26-10591-t002:** Linear regression results for the association between *APOE* alleles and changes in Z-scores from baseline to follow-up for cognitive domains.

	Visuospatial Skills/Speed					Verbal Memory					Verbal Fluency				
TOTAL	n	β	95% CI	*p*	*p*-perm	n	β	95% CI	*p*	*p*-perm	n	β	95% CI	*p*	*p*-perm
*APOE* ε4	358	0.235	−0.099/0.569	0.169	0.339	374	−0.147	−0.320/0.027	0.098	0.320	372	−0.168	−0.343/0.007	0.060	0.222
*APOE* ε2	358	0.026	−0.436/0.487	0.914	1	374	−0.238	−0.474/−0.001	0.049	0.180	372	0.105	−0.137/0.348	0.395	0.840
MEN	n	β	95% CI	*p*	*p*-perm	n	β	95% CI	*p*	*p*-perm	n	β	95% CI	*p*	*p*-perm
*APOE* ε4	225	0.306	−0.176/0.778	0.214	0.377	233	−0.163	−0.352/0.026	0.093	0.270	233	−0.192	−0.392/0.007	0.060	0.217
*APOE* ε2	225	−0.007	−0.768/0.753	0.984	1	233	−0.382	−0.670/−0.095	0.010	0.027	233	0.135	−0.176/0.446	0.395	0.828
WOMEN	n	β	95% CI	*p*	*p*-perm	n	β	95% CI	*p*	*p*-perm	n	β	95% CI	*p*	*p*-perm
*APOE* ε4	133	−0.011	−0.185/0.163	0.902	1	141	−0.116	−0.497/0.266	0.552	0.961	139	−0.075	−0.440/0.290	0.686	0.987
*APOE* ε2	133	0.100	−0.086/0.287	0.293	0.714	141	−0.017	−0.448/0.414	0.939	1	139	0.041	−0.362/0.445	0.841	1

Note: n represents the actual number of participants with complete data for all genotypes, cognitive assessments, and covariates included in the model. β coefficients (and 95% CI) represent the effect of each extra minor allele. Age, sex, years of education, baseline depression, baseline REGICOR, and baseline domain Z-score were included as covariates in all analyses. Analyses were conducted under an additive genetic model (i.e., allele-dose-dependent). *p*-perm: probability of the observed *p*-values after 1000 permutations.

**Table 3 ijms-26-10591-t003:** Linear regression results for the association between *APOE* alleles and Z-scores at follow-up for cognitive domains.

		Visuospatial Skills/Speed					Verbal Memory					Verbal Fluency			
TOTAL	n	β	95% CI	*p*	*p*-perm	n	β	95% CI	*p*	*p*-perm	n	β	95% CI	*p*	*p*-perm
*APOE* ε4	358	0.267	−0.068/0.602	0.119	0.273	374	−0.180	−0.391/0.030	0.094	0.323	372	−0.178	−0.396/0.040	0.111	0.340
*APOE* ε2	358	−0.020	−0.483/0.443	0.933	1	374	−0.028	−0.313/0.258	0.850	0.999	372	0.283	−0.016/0.582	0.064	0.212
MEN	n	β	95% CI	*p*	*p*-perm	n	β	95% CI	*p*	*p*-perm	n	β	95% CI	*p*	*p*-perm
*APOE* ε4	225	0.333	−0.149/0.816	0.177	0.331	233	−0.188	−0.424/0.047	0.118	0.367	233	−0.224	−0.483/0.035	0.092	0.318
*APOE* ε2	225	−0.112	−0.861/0.642	0.772	0.996	233	−0.354	−0.714/0.007	0.056	0.199	233	0.285	−0.116/0.685	0.165	0.488
WOMEN	n	β	95% CI	*p*	*p*-perm	n	β	95% CI	*p*	*p*-perm	n	β	95% CI	*p*	*p*-perm
*APOE* ε4	133	0.031	−0.155/0.218	0.741	0.999	141	−0.168	−0.614/0.278	0.462	0.889	139	−0.034	−0.459/0.392	0.876	1
*APOE* ε2	133	0.128	−0.072/0.329	0.213	0.624	141	0.437	−0.037/0.911	0.073	0.247	139	0.287	−0.174/0.748	0.224	0.590

Note: β coefficients (and 95% CI) represent the effect of each extra minor allele. Age, sex, years of education, baseline depression, and baseline REGICOR were included as covariates in all analyses. Analyses were conducted under an additive genetic model (i.e., allele-dose-dependent). *p*-perm: probability of the observed *p*-values after 1000 permutations.

## Data Availability

The data presented in this study are available on request from the corresponding author due to privacy and ethical restrictions.

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
