# Peer review of "Sex and APOE ε2 Interactive Effects on the Longitudinal Change in Cognition in a Population-Based Cohort of Older Adults with Vascular Risk Factors"

_ijms, 2025, doi:10.3390/ijms262110591_

Round 1
Reviewer 1 Report
Comments and Suggestions for Authors
The manuscript presents a longitudinal study about the interactive effects of APOε2 genotype and sex on cognitive decline over a 7-year period in older adults with vascular risk factors. I consider that the topic is highly relevant and the fact that this is a longitudinal study is a strength.
However there are several aspects that I think need improvement.
I think there is a need to be more detalied about the material and the methods used. In the results section, figure 1 has the label of inclusion and exclusion criteria, but I think it is only just a flowchart of the study, not an actual inclusion or exclusion criteria.
The loss of participants between baseline and follow-up raises concerns. I think that the impact of the dropout on genotype representation and baseline cognitive performance should be more explicitly discussed (maybe a sensitivity analysis would help).
The paper discusses prior cross-sectional findings where APOε2 had a protective effect in women, but longitudinal data suggests a lower effect in men and no effect in women. Although RTM is proposed as a partial explanation, this issue deserves a clearer narrative and graphical support.
Also, it would be useful a clearer justification for initially including BDNF based on prior evidence.
I would also propose a discussion about the limitations of the study based on the lack of the lipid data at follow-up.
The abstract is slightly dense - maybe the last 2 sentences could be more clear.
The references are comprehensive and relevant, but could be included studies after 2020 on sex-genotype interactions to add strength.
Comments on the Quality of English Language
Some sentences are long or repeat similar ideas (for example - “...participants were cognitively reassessed between April 2016 and May 2017, after a median of 7.16 [6.91–7.75] years.”
There is some awkward phrasing: This kind of research is warranted to further understand the mechanisms involved in the protective and deleterious effects...
Also, in some places the paper jumps from past and present tense.
Reviewer 2 Report
Comments and Suggestions for Authors
General comment:
This is an interesting study that investigates the longitudinal impact of APOE ε2 allele in relation to sex on cognitive change in older adults with vascular risk. The authors use a large, well-characterized cohort with a 7-year follow-up and appropriate statistical modeling (including control for regression to the mean, permutation testing, and interactions). The study contributes new insights into the sex-specific effects of APOE ε2 on cognition over time, especially its potential detrimental role in males whic is a less studied topic. However, there are a few areas where the manuscript could be strengthened.
Introduction
The introduction sets an expectation of a protective effect of APOE ε2, particularly in women. However, the actual findings are mixed or contrary to this hypothesis (no effect in women, detrimental effect in men). It would better to revise the introduction and abstract to reflect that the directionality of ε2 effects remains debated, with emerging evidence for heterogeneity.
Materials and methods
Clarify the REGICOR score
The manuscript uses the REGICOR score to adjust for cardiovascular risk, but does not explain what this score represents. Please include a brief explanation in the Methods section, noting that REGICOR is a recalibrated version of the Framingham coronary risk function for the Spanish population. It's extremely important to clarify which variables are included since it would help readers unfamiliar with this score understand its role as a covariate in the cognitive models.
Consideration of mixed-effects modeling
The current analysis is based on change scores (follow-up minus baseline) modeled via linear regression. While this is acceptable, the use of mixed-effects models (e.g., random intercept models) could better account for within-subject correlation, partial missingness, and interindividual variability. If feasible, the authors may consider implementing such models in or clarifying why this approach was not used.
Sensitivity analysis excluding participants with clinically relevant depressive symptoms
Although depressive symptoms (GDS-15) were included as a covariate, no sensitivity analysis was performed excluding participants with GDS >5, a commonly used threshold for clinically significant depression. Such an analysis could help determine whether the genetic associations observed are robust or potentially confounded by subthreshold affective symptoms, particularly in verbal memory performance.
Discussion
BDNF and APOE interaction
BDNF showed no significant main effects on cognitive change and only marginal interactions with APOE . The authors should consider expanding the discussion to clarify:
-
Whether the inclusion of BDNF was hypothesis-driven (e.g., based on prior evidence of gene–gene modulation in neuroplasticity and cognition), or purely exploratory.
-
The biological rationale for expecting an APOE×BDNF interaction (e.g., their shared roles in synaptic plasticity, neuroprotection, and cognitive aging).
-
Whether the lack of significant findings could be due to limited power, the small proportion of Met allele carriers, or differences in cognitive domains assessed compared to prior studies.
Limitations
The authors may consider acknowledging, in the Limitations section, that the study did not include potentially relevant environmental exposures, such as air pollution, which have been shown to influence cognitive trajectories through endocrine and inflammatory pathways. For instance, pollutants like PM10 and PM2.5 are known to affect key mediators such as cortisol and TNF-α, both of which are implicated in cognitive decline and neurodegeneration.
Please see: Dolcini J. et al., “Association between TNF-α, cortisol levels, and exposure to PM10 and PM2.5: a pilot study,” Environ Sci Pollut Res, 2024. doi:10.1007/s11356-024-30994-2.
